# Organic Cation Transporters in the Lung—Current and Emerging (Patho)Physiological and Pharmacological Concepts

**DOI:** 10.3390/ijms21239168

**Published:** 2020-12-01

**Authors:** Mohammed Ali Selo, Johannes A. Sake, Carsten Ehrhardt, Johanna J. Salomon

**Affiliations:** 1School of Pharmacy and Pharmaceutical Sciences and Trinity Biomedical Sciences Institute (TBSI), Trinity College Dublin, Dublin 2, Ireland; jsake@tcd.ie (J.A.S.); ehrhardc@tcd.ie (C.E.); 2Faculty of Pharmacy, University of Kufa, 540011 Al-Najaf, Iraq; 3Translational Lung Research Center Heidelberg (TLRC), German Center for Lung Research (DZL), Department of Translational Pulmonology, University of Heidelberg, 69120 Heidelberg, Germany

**Keywords:** pulmonary drug delivery, SLC22A1–5, lung epithelium, drug uptake, β2-agonists, chronic lung diseases, anticholinergics

## Abstract

Organic cation transporters (OCT) 1, 2 and 3 and novel organic cation transporters (OCTN) 1 and 2 of the solute carrier 22 (SLC22) family are involved in the cellular transport of endogenous compounds such as neurotransmitters, l-carnitine and ergothioneine. OCT/Ns have also been implicated in the transport of xenobiotics across various biological barriers, for example biguanides and histamine receptor antagonists. In addition, several drugs used in the treatment of respiratory disorders are cations at physiological pH and potential substrates of OCT/Ns. OCT/Ns may also be associated with the development of chronic lung diseases such as allergic asthma and chronic obstructive pulmonary disease (COPD) and, thus, are possible new drug targets. As part of the Special Issue “Physiology, Biochemistry and Pharmacology of Transporters for Organic Cations”, this review provides an overview of recent findings on the (patho)physiological and pharmacological functions of organic cation transporters in the lung.

## 1. Introduction

Organic cation transmembrane transporters belonging to the solute carrier family 22 (i.e., SLC22A1–A5) are increasingly recognised as “impactors” of drug disposition in the respiratory tract [1,2,3]. SLC22A1–A5 transporters can be further divided according to the driving force of cation transport into membrane-potential-sensitive organic cation transporters (OCTs) or Na^+^ and pH-dependent novel organic cation transporters (OCTNs) [4,5]. The human OCT subclass consists of OCT1 (SLC22A1), OCT2 (SLC22A2) and OCT3 (SLC22A3), whereas the OCTN subclass includes OCTN1 (SLC22A4) and OCTN2 (SLC22A5). According to the human protein atlas, hOCT1 is a 61 kDa protein exhibiting a broad tissue distribution with high expression levels in the liver [6,7]. hOCT2, with a less ubiquitous expression pattern than hOCT1 and hOCT3, is most strongly expressed in the kidneys with an approximate molecular size of 63 kDa [6,8]. The tissue expression pattern of OCT3 (61 kDa) is very broad, with high levels being observed in the liver, skeletal muscle, placenta and heart [8,9]. OCTN1 and OCTN2 also exhibit broad tissue distribution, and both are approximately 62 kDa proteins [8,9]. OCT/N transporters participate in the cellular transport of a broad spectrum of endogenous and exogenous organic cations and zwitterions such as neurotransmitters and xenobiotics. Thus, they are involved in clinical drug–transporter interactions and at the same time, they perform important physiological functions [9,10]. OCT1, for example mediates thiamine uptake, modulates hepatic glucose and lipid metabolism and thereby plays a role in hepatic steatosis [11,12]. In the brain, OCT2 and OCT3 are involved in the regulation of a variety of normal central nervous system functions related to mood as well as salt-intake behaviour and osmoregulation [13,14,15]. Detailed information on OCTs’ expression and function in organs (except for the lung) was recently reviewed and can be found in this Special Issue of International Journal of Molecular Sciences [16]. With a focus on the lung, reviews by Salomon et al. [2] and Nickel et al. [3] have provided comprehensive overviews of pulmonary OCT/Ns’ expression and function to the reader. Generally, OCTs in airway epithelial and smooth muscle cells accept physiological substrates such as dopamine, histamine, serotonin and acetylcholine. OCTN1 and OCTN2 mediate the uptake of ergothioneine (ESH) and l-carnitine, respectively. The exact roles of OCT/N transporters in lung (patho)physiology and pharmacology, however, are still not fully understood, and these topics are discussed in this review.

## 2. Expression and Subcellular Localisation of OCT/Ns in Lung-Derived Cell Lines, Pulmonary Cell Cultures and Lung Tissues in Health and Disease

The expression of OCT/Ns in the lung has been studied in several cell lines of human respiratory epithelial origin (e.g., A549, NCl-H441, BEAS-2B, Calu-3 and 16HBE14o-), in primary airway epithelial cells and in lung tissues on the mRNA and protein level [17,18,19,20,21,22,23], and has been reviewed comprehensively in our pervious publication [3]. There is some consensus that OCT1, OCT3, OCTN1 and OCTN2 are found ubiquitously throughout the lung epithelium. OCT2 expression is more controversial. The transporter was absent in many of lung-derived cell lines [17,18,23,24,25,26], except for NCl-H441 [20,27]. In the case of primary cultures of human tracheal, bronchial and alveolar epithelial cells and in human whole-lung tissue, a lot of conflicting data have been published [17,19,21,23,25,28,29].

The regional expression and subcellular localisation of OCT/Ns in the airways remains, to some extent, elusive. Immunohistochemistry (IHC) studies carried out in human bronchi revealed positive OCT1 and OCT2 staining in the apical membrane of ciliated epithelial cells, intracellular OCT1 staining in ciliated epithelial cells, and OCT3 staining in the basolateral membrane of intermediate cells and the entire plasma membrane of basal cells [28]. OCT3 transcript and protein expression were confirmed in primary human bronchial and vascular smooth muscle cells [30]. Positive OCTN1 and OCTN2 stainings were observed on the apical and the lateral membranes of human primary bronchial epithelial cells [31]. OCTN1 showed the strongest expression of all OCT/Ns in bronchi but was found to be expressed at a lower degree in peripheral lung tissue [32]. Data are still scarce on cell-type-specific expression of OCT/Ns, and hence it is rather difficult to identify the physiological function of each transporter protein. We highlighted this issue in another review in 2016 [3], however, the field has not advanced significantly since then.

A limited number of studies have been published connecting OCT/N expression to lung diseases. Genome-wide association studies (GWAS) have identified *SLC22A5* variants being linked to primary systemic carnitine deficiency and asthma [33,34]. Furthermore, due to its potential role in histamine release and clearance in the airways, an association between *SLC22A3* gene polymorphisms and the severity of asthma has been proposed [35,36]. No differences in mRNA expression levels of OCT1, OCT3, OCTN1 and OCTN2 between ex-smokers with a severe stage of COPD and healthy subjects were reported in a study by Berg and colleagues [32]. Subsequently, the same group’s IHC analysis confirmed OCTN1 and OCTN2 expression in the epithelial cells of the bronchi, bronchioles and alveolar type II epithelial cells as well as in alveolar macrophages. However, no differences between COPD and healthy subjects were visible in IHC [31]. It should be noted that neither was immunoblot analysis carried out nor were IHC signal intensities measured in order to quantify potential differences in protein expression between the two groups. In another study, Calu-3 cells were grown for 21 days under air-interfaced culture (AIC) conditions and exposed to pro-inflammatory lipopolysaccharide (LPS) or house dust mite extract (HDM) to simulate asthmatic-like conditions at the epithelium in vitro [37]. The LPS challenge significantly upregulated the expression of OCT1, OCT3, OCTN1 and OCTN2 on mRNA and protein levels. HDM had similar effects on OCT1, OCT3 and OCTN2 mRNA and protein expression. However, when Calu-3 cells grown for the same duration and under similar conditions were exposed to LPS for shorter periods of time, other researchers did not observe any change in mRNA levels of OCT1, OCT3 [24] or OCTN2 [38]. Stimulation with tumour necrosis factor-α (TNF-α) and the Th2-cytokine, IL-4 also did not result in any change in OCT1, OCT3 [24] and OCTN2 mRNA levels [38]. Regretfully, in neither study, were mRNA expression data supported by protein expression analysis. In contrast, when alveolar A549 cells were exposed to similar LPS concentrations, mRNA and protein levels of OCTN1 and OCTN2 were downregulated, which was accompanied by a significant reduction in the uptake of the model substrate 4-[4-(dimethylamino)-styryl]-N-methylpyridinium (ASP^+^) [39]. Likewise, challenging A549 cells with cigarette smoke extract (CSE), LPS or both caused a significant reduction in OCTN1 and OCTN2 mRNA expression levels [40]. It has been demonstrated that A549 and BEAS-2B cells respond distinctly, in terms of cytokine release, to LPS stimulation [41]. The latter may suggest different pathways involved in alveolar and bronchial epithelial cells upon LPS challenge and, together with different LPS exposure times used in the above-mentioned studies, may explain the discrepancies in OCT/N expression regulation observed in response to LPS stimulation in Calu-3 and A549 cells. Overall, some promising first results suggest that there indeed could be differences in OCT/N expression (and function) in healthy vs. diseased lungs, which needs to be further investigated using appropriate in vitro and experimental animal models as well as lung tissue specimens from patients with respiratory diseases.

OCT/N expression and subcellular localisation in the lungs need to be conclusively studied. For example, cell surface protein biotinylation should be performed to obtain clear-cut subcellular localisation data for OCT/Ns. However, it is essential to use high-quality antibodies, which might have to be generated first. Proteomics profiling will help not only to quantify expression levels of OCT/Ns but also to look into the association between molecular pathways of OCT/Ns in the lung, in health and disease [42]. Lastly, single-cell analysis approaches will allow to sort cell populations of the lung and to determine the expression of OCT/Ns in individual cells (i.e., epithelial vs. immune cells). In this context, first evidence was given that alveolar macrophages also express OCTN1 and OCTN2 [31].

## 3. OCT/Ns (SLC22A1–A5) Transporter Function in Lung Physiology and Pathophysiology

Despite being involved in the transport of essential endogenous substrates (Figure 1), little information is available about the role of OCT/Ns in physiological functions and under pathological conditions in the lung. Uncovering (patho)physiological functions of OCT/Ns in the lung is a key step in developing new drug therapies for respiratory disorders [10,43]. Molecular investigations are necessary to further identify and validate potential endogenous substrates of OCT/Ns. Transport studies of substrate candidates are traditionally performed utilising radioactive isotopes or liquid chromatography–mass spectrometry. Potentially, a novel assay that detects shifts in thermostability of the transporter protein in the presence of specific substrates may be employed to detect transporter–substrate interactions of lung OCT/Ns [44]. The main advantage of this assay is that it allows large-scale screening to identify candidate substrates from libraries of unlabelled compounds. However, the assay cannot discriminate between substrates, inhibitors or competitors and requires purified proteins. Novel approaches such as transporter tandems might also help to assess the influence of gene polymorphisms on OCT/Ns functional activity and to determine their (patho)physiological consequences in the lung [45] (see Section 3.5 for further details).

### 3.1. OCT1, 2 and 3

OCTs transport a wide range of physiologically important endogenous substrates, including hormones such as prostaglandin E_2_ and neurotransmitters such as acetylcholine (ACh), dopamine, serotonin, epinephrine, norepinephrine and histamine [3,6,46,47]. In the lung, the current knowledge of OCTs in these molecular interplays is primarily limited to non-neuronal ACh and serotonin transport. OCT1/Oct1 and OCT2/Oct2 have been demonstrated to be involved in the luminal release of non-neuronal ACh in human airway epithelial cells in vitro [28] and in mice in vivo [48]. ACh has various physiological functions in the airway. It regulates mucus secretion and clearance, bronchoconstriction, histamine release from mast cells [49,50,51,52,53] and promotes airway remodelling, particularly during inflammatory lung diseases such as asthma and COPD [51,54]. ACh has also been reported to have a negative impact on the progression of human lung cancer by acting as a growth factor [55]. In the case of serotonin, suppression of OCT3 by corticosterone has been reported to block serotonin-induced bronchoconstriction in mice [48].

Recent insights underline the importance of the above-mentioned neurotransmitters in pulmonary (patho)physiology. A study carried out in mice linked dopamine to a higher susceptibility of children to asthma compared to that of adults [56]. The study showed that sympathetic nerves innervating young mice lungs primarily produce dopamine, which triggers inflammatory reactions related to asthma [56]. Similarly, histamine is a bronchoconstrictor and plays a role in the pathogenesis of asthma and a number of other allergic disorders [36,57,58]. In addition to its bronchoconstricting action, higher plasma levels of serotonin have been observed in asthmatic [59] and COPD patients [60]. Despite having their own high-affinity physiological transporters (e.g., SLC6A3), alteration in OCT activity has been reported to influence the homoeostasis of these neurotransmitters and may result in pathophysiological consequences in the lungs. For example, knocking out of OCT2 and OCT3 in mice resulted in mood abnormality by interfering with monoamine neurotransmitters’ clearance in the brain [13,61].

OCTs are also involved in interactions between therapeutic drugs with endogenous substrates. In this context, a number of commonly prescribed drugs have been shown to interfere with the uptake of histamine and monoamine neurotransmitters (i.e., serotonin, epinephrine, norepinephrine and dopamine) into OCT1-overexpressing HEK-293 cells and primary human hepatocytes [46,62]. However, such potential interactions have not yet been investigated in cells of lung origin. Answering further questions on physiological function, therefore, remains challenging. As mentioned previously, OCT2 is either absent or expressed at low levels in human airway epithelium. Thus, the focus shall be on OCT1 (>OCT3) to assess whether alterations in the expression and/or activity of these transporters result in pathophysiological consequences due to changes in the pulmonary disposition of the above-mentioned endogenous substrates. In particular, increased OCT1 and/or OCT3 activity can enhance dopamine uptake into respiratory epithelial cells, which is subsequently metabolised intracellularly by histamine *N*-methyltransferase [36,63], resulting in reduced neurotransmitter concentration in the extracellular airways space, which may have a positive impact on childhood asthma and/or vice versa.

### 3.2. OCTN1

OCTN1 facilitates the cellular uptake of its physiological substrate, ESH [64]. Gründemann and colleagues have demonstrated that the transporter is highly specific for ESH and hence the name “ergothioneine transporter” (ETT) has been proposed instead of OCTN1 [65,66]. Expression of OCTN1 in distinct tissues can, therefore, serve as a specific molecular indicator of intracellular ESH activity [65,66]. According to a number of in vitro and in vivo studies, ESH acts as a powerful free-radical scavenger and can modulate inflammation [67,68,69,70]. Polymorphisms in the *SLC22A4* gene have been linked with susceptibility to many chronic inflammatory diseases such as ulcerative colitis and Crohn’s disease [71,72,73] as well as to tooth loss and adiposity in women [74]. In the lung, a protective role for ESH against inflammatory and oxidative-stress-induced damage has been demonstrated. For instance, pre-treatment of alveolar epithelial A549 cells with ESH inhibited TNF-α and H_2_O_2_-mediated IL-8 release and activation of nuclear factor kappa B (NF-κB) [75]. ESH was, therefore, proposed as a potential therapeutic intervention for chronic inflammatory pulmonary disorders such as COPD [76,77]. In a rat model of acute respiratory distress syndrome, intravenous ESH treatment before and after cytokine insufflation attenuated acute lung injury and inflammation [78]. The authors, however, did not carry out detailed histopathological studies to support their results. The physiologic transporter of ESH (i.e., OCTN1) itself was not considered in these studies. For drawing conclusions on the clinical impact, it is not only essential to investigate the substrate itself but also potential links between these respiratory disorders and reduced OCTN1 activity, e.g., through genetic polymorphisms or drug–drug interactions.

### 3.3. OCTN2

OCTN2 mediates the uptake of l-carnitine [6,79] and hence controls its homoeostasis, which is achieved by endogenous biosynthesis (mainly in the liver and kidney), intake from the diet and renal reabsorption [80]. l-carnitine is a highly polar zwitterionic and naturally occurring compound that plays a key physiological role in the mitochondrial β-oxidation of fatty acids and, consequently, in cellular energy production [80,81]. The process of fatty acid oxidation involves the consumption of large amounts of oxygen, which are reduced to H_2_O, resulting in reduction in intracellular reactive oxygen species formation. l-carnitine also regulates the activity of several enzymes involved in protection against oxidative-stress-induced damage [82]. Based on in vitro and in vivo studies, l-carnitine can scavenge free radicals and counteract oxidants such as peroxyl radicals, hydrogen peroxide and peroxynitrite [83,84,85,86,87]. Reduced intracellular l-carnitine levels in turn impair cellular energy generation and result in deterioration of the function of several organs [80].

The clinical relevance of the substrate l-carnitine in lung (patho)physiology is well investigated. Studies have demonstrated that reduction in oxidative stress levels and increased activities of many antioxidant enzymes following administration of l-carnitine supplements are seen in healthy volunteers [88], suggesting a protective role against oxidative-stress-induced damage [89]. Moreover, lower serum l-carnitine levels were found in children with moderate persistent asthma compared to those in healthy volunteers, and six months of l-carnitine supplementation to the asthmatic children improved childhood-asthma control test and pulmonary function test parameters [90]. A significant reduction in serum l-carnitine level was observed in asthmatic children during acute exacerbations and shortly thereafter. The difference in l-carnitine levels between the groups of healthy and asthmatic children, however, was not significant [91]. These data should be carefully considered due to low numbers of patients and healthy controls included in the above-mentioned studies. l-carnitine also plays a role in pulmonary surfactant synthesis. In murine alveolar epithelium, it was proposed that when fatty acid oxidation is impaired, pulmonary surfactant levels and lung function are decreased [92]. Acylcarnitines, which are catabolised from carnitine, directly inhibit the activity of alveolar surfactant [92]. Furthermore, l-carnitine has been proposed to play a role in respiratory distress syndrome (RDS). Treatment of new-borns with RDS with l-carnitine has been demonstrated to reduce surfactant requirement, to shorten the duration of mechanical ventilation needed and to reduce the incidence of bronchopulmonary dysplasia [93]. These clinical findings, however, were not yet linked to the expression and function of OCTN2. In the future, these data could permit an identification of a clinically relevant dysregulation or dysfunction of the physiological transporter in these respiratory disorders. Lately, OCTN2-mediated l-carnitine uptake, via maintaining energy supply, was discussed to play a role in sustaining respiratory ciliary beating and consequently, airway mucociliary clearance, by which excess mucus and potentially harmful foreign particles are removed from the airways [94].

Due to the physiological importance of OCTN2 in regulating l-carnitine homoeostasis, data on expression and function in different preclinical models is steadily increasing. In vitro studies confirmed a cellular OCTN2-mediated accumulation of l-carnitine and acetyl-l-carnitine in human respiratory epithelial cell models [22,95,96]. However, when acetyl-l-carnitine uptake into a number of respiratory epithelial cell lines (i.e., NCl-H441, A549 and Calu-3 cells) was compared to that in primary alveolar epithelial cells, the kinetics and inhibitor specificities were significantly different, similar to the functions of OCTs (see Section 3.4) [22]. In vivo studies revealed an active accumulation of l-carnitine in mice trachea without a systemic absorption [97]. This was supported by studies in an intact isolated, perfused rat-lung model, where data revealed no carnitine-sensitive pulmonary absorption [95].

### 3.4. Functional Studies Using Exogenous OCT/N Substrates

The uptake and transport of xenobiotics has been reviewed before [98]. To measure OCT/Ns’ function, the fluorescent dye ASP^+^ [19,27,98,99] has often been used. ASP^+^, however, lacks selectivity as an OCT substrate as it is also a substrate for a number of plasma membrane neurotransmitter transporters [100,101]. Tetraethylammonium (TEA), 1-methyl-4-phenylpyridinium (MPP^+^) and decynium-22 [6] have been used mainly in in vitro studies and are more specific for OCTs but lack selectivity for the individual OCT subtypes [18,102]. Distinct TEA uptake kinetics and patterns of OCT inhibitory effects were observed in NCl-H441, A549 and Calu-3 cell lines when compared to those in primary alveolar epithelial cells [102]. These results point out the relevance of a physiological epithelial cell model. When MPP^+^ uptake was studied, only the above-mentioned respiratory epithelial cell lines were used [18]. The authors suggested differential contributions of OCTs to the MPP^+^ uptake [18]. Thus, these results have to be carefully discussed in terms of physiological propagation. To our knowledge, no other data on pulmonary OCT/N-mediated drug transport was published since our last review in 2016 [3]. It is still a rather difficult assignment to conclusively determine the regional OCT/N transporter activity in the lung. The affinities of OCTs for exogenous and endogenous substrates have been comprehensively summarised in several reviews [6,9,103] with a focus on the liver and the kidneys. Studies of clinically relevant OCT-involving drug–drug interactions have recently been reviewed by Koepsell [103,104]. In the lung, very few pharmacokinetic studies have been performed [18,102,105]. Interactions of beta-agonists, corticosteroids and anticholinergics may occur in the lung, and we have focussed on recent data in Section 4; however these studies are mainly based on interaction of drugs and model substrates.

### 3.5. Relation of OCT/N Expression, Subcellular Localisation and Function

Some of the reported discrepancies in OCT/N expression levels mentioned in Section 2 are likely due to inconsistencies in passage numbers, time in culture, culture conditions (i.e., AIC vs. liquid-covered conditions (LCC)) and medium supplements. For example, OCT3 transcripts were detected in Calu-3 cells grown for 21, but not for 14, days [25]. Higher OCTN1 and OCTN2 mRNA levels were observed in 16HBE14o- cells cultured for 18 days than in those grown for 11 days [17]. Culturing Caco-2 cells in media supplemented with high glucose concentrations has been shown to reduce OCT3 activity and mRNA and protein expression levels [106]. Likewise, cell culture conditions and time in culture have been demonstrated to have a significant impact on the mRNA expression levels of a number of SLC transporters in Calu-3 [107] and primary human nasal epithelial cells [108]. Method standardisation is, therefore, crucial for future studies in order to validate epithelial cell models and freshly isolated human lung epithelial cells as surrogates for in vivo studies. Furthermore, a lack of high-quality antibodies has been a major challenge in OCT/N research. In particular, when using commercially available antibodies for protein detection via confocal laser scanning microscopy, the localisation of the observed signals was not always unambiguous (see below). This also constrains investigations in novel in vitro cell systems, such as 3D epithelial cell models, to evaluate their suitability as models for OCT/N research. For further reading on pulmonary epithelial cell models, please refer to the following reviews and publications [109,110,111,112].

Investigating the activity of a transporter protein can be more challenging than studying variations in its mRNA or gene product expression. Much of lung OCT/N research is solely based on expression analysis [17,23,113,114] and hence presents a strongly unilateral view on OCT/Ns. Expression studies using immunoblotting or liquid chromatography–tandem mass spectrometry (LC-MS/MS) may not precisely foretell a transporter’s activity due to the detection of inactive intracellular transporter protein. LC-MS/MS has been used to study the expression levels of a number of transporters, including OCT/Ns, in the plasma membranes fractions isolated, via gradient centrifugation, from human whole-lung tissues, primary cells and a number of lung-derived cell lines [20,21,29]. The purity of plasma membrane fractions, however, could not be confirmed. Transporter tandems has been recently proposed as a precise tool to determine a transporter’s activity in the plasma membrane [45]. In short, a link is created between two transporters by joining their cDNAs in a single open reading frame, resulting in a 1:1 stoichiometry and, thereby, enabling to measure the activity of one transporter by the activity of the second one. For example, OCTN1–OCT2 tandems were used to assess the activity of OCTN1 [45]. Moreover, linking OCTN1 as a reference transporter with a number of SLC and SLCO transporters (i.e., OCT1, OCT2, OAT1, OAT3, OATP1B1 and OATP1B3), has demonstrated that transporter tandems can be a useful tool to assess the activity of transporters belonging to other families [45]. However, there must be no functional overlap between the reference and the test transporter of the tandem, which must be verified by uptake experiments with the individual unconnected transporters. In addition, suitable linker peptides should be used between the two transporters. The sequence and the length of the linkers as well as the order of transporters in the tandems may have a strong impact on transporters’ expression and activity and, therefore, must be carefully validated [45].

Activity studies using ASP^+^ suggested OCT/N localisation on the apical side of Calu-3 cell monolayers grown under AIC conditions for 21 days [25]. IHC analysis, which confirmed the expression of OCT1, OCT3 and OCTN2, however, was inconclusive regarding the subcellular localisation of the transporters [25]. In contrast, bidirectional MPP^+^ transport across Calu-3 monolayers grown under similar conditions and for similar duration was proposed to be mediated by both OCT1 and OCT3 at the apical and OCT3 at the basolateral side of the polarised monolayers [18]. In the same study, transport studies carried out in NCI-H441 and A549 cells suggested apical expression of OCT1 and OCT3 [18]. The authors, however, could not support their data with proper subcellular localisation experiments. More recently, MPP^+^ uptake studies, indicated that OCT1 and OCT3 are active on the basolateral membrane of the 3D cell model composed of normal human bronchial epithelial cells [24]. l-carnitine transport studies suggested OCTN2 to be active on the basolateral side of Calu-3 cells and the 3D cell model [38]. IHC analysis was suggestive of a basolateral localisation of the transporter in Calu-3 monolayers, but the signals were inconclusive in the 3D cell model [38].

Taken together, these data suggest that functional activity studies may fail to accurately predict the subcellular localisation and/or expression of OCT/Ns and vice versa. Thus, monitoring expression and function of OCT/Ns simultaneously is necessary to define active transporter proteins and to determine the source of published discrepancies in either expression or functional analysis.

## 4. Pharmacological Aspects of OCT/N Transporters in the Lung

### 4.1. Interaction of OCT/N Transporters with Inhaled Drugs

Inhaled bronchodilator drugs must pass through the airway epithelial barrier to reach their target receptors in the underlying airway smooth muscle cells. Many of these drugs belonging to the muscarinic receptor antagonists and β_2_-agonists are cations and are positively charged at physiological pH-values of the lung lining fluid. Thus, OCT/Ns may play a potential role in their pulmonary disposition, pharmacokinetics, safety and efficacy profile. The involvement of OCT/N transporters in interactions with inhaled drugs has been intensively discussed in previous reviews [2,3].

The role of OCT/Ns in the pulmonary disposition of inhaled drugs is an ongoing research topic; however, data are mainly limited to in vitro studies. Despite being essential for initial screenings of inhaled drugs’ interactions with membrane transporters, in vitro studies are insufficient to confirm the clinical significances of such interactions, and a number of challenges and disadvantages are associated with them. First, an ideal in vitro lung epithelial model for drug disposition studies still does not exist. Second, the different cell composition of proximal and distal lung epithelium may result in different OCT/N expression and activity profiles [17,98]. Third, determination of inhaled drug concentrations in the epithelial lining fluid following drug inhalation is extremely difficult because of the complex anatomical nature of the lung [25,115]. Thus, the concentrations of inhaled drugs applied in in vitro studies to assess their interaction with OCT/N transporters may be clinically irrelevant. The concentration of drugs used in in vitro studies is of particular importance because it influences the ratio of passive membrane diffusion and transporter-mediated uptake [9]. Generally, when high concentrations are used, passive membrane permeation predominates resulting in an underestimation of the transporter impact [9]. In vitro data may, therefore, not predict the in vivo airway-to-blood absorption process accurately and should be carefully considered. Validation can be achieved by ex vivo, in vivo and clinical studies [10]. Recent reviews have covered preclinical models (in vitro, ex vivo and in vivo) that are currently implemented in pulmonary drug delivery studies [111,112]. Moreover, new experimental in silico models such as Mimetikos Preludium™ (Emmace Consulting) and SimCyp Simulator™ (Certara) can be used to estimate the regional absorption of inhaled drugs [116,117]. It is necessary to point out that such in silico models may fail to precisely describe the pulmonary pharmacokinetics of drugs because the validity of the simulation depends on the quality (and quantity) of data used to inform the model. These data, however, are scarce and were generated under different experimental conditions. Moreover, none of these models has been specifically designed for use in pulmonary drug delivery. Nonetheless, in silico models might prove useful to assess the impact of OCT/Ns in pulmonary disposition of inhaled drugs, once sufficient and reliable in vitro and in vivo data have been generated to inform them.

Many questions related to the pharmacological roles of OCT/Ns in the lung have been raised and remain open. Which member of the family is of particular importance? Are OCT/N–drug interactions clinically relevant, and do they influence drug efficacy and toxicity?

#### 4.1.1. Interaction with β_2_-Agonists

Ehrhardt et al. were the first to propose the involvement OCT/Ns in the active transport of salbutamol across monolayers of human bronchial epithelial cell lines in vitro [118]. In another study, salbutamol was suggested to be a specific substrate and inhibitor of OCT1 in human distal respiratory epithelial cells [105]. Horvath and colleagues observed that OCTN2 function can be inhibited by salbutamol and formoterol in normal human bronchial epithelial (NHBE) cells in vitro [19]. Whilst they found no role for OCT/Ns in the transepithelial transport of salbutamol across the cell monolayers, paracellular diffusion was suggested to be the predominant mediator of the inhaled drug translocation across lung mucosa [119]. Salbutamol, formoterol and ipratropium bromide have also been demonstrated to inhibit OCT1, OCT3, OCTN1 and OCTN2 in bronchial epithelial Calu-3 cells [25]. A number of nonsteroidal anti-inflammatory drugs interfered with salbutamol uptake into Calu-3 cells via a mechanism involving the inhibition of OCT transporters [120]. In a more recent study, data showed that epithelial asthmatic-like challenges can enhance the transepithelial permeability of salbutamol through OCT transporters’ overexpression in vitro [37]. A delay in the pulmonary absorption of salbutamol and GW597901, a long-acting β_2_-agonist, was observed in isolated human lung reperfusion model after nebulisation of l-carnitine via a mechanism involving competition with the OCTN2 transporter [121]. However, the authors could not determine the duration of delay due to the limited viability time of the model.

Fenoterol, a short-acting β_2_-agonist, has been identified as a substrate of OCT1 [122,123]. When a number of heritable OCT1 variants were overexpressed in HEK-293 cells, the uptake of this drug was completely abolished or substantially reduced [123]. Clinically, following intravenous administration of fenoterol to healthy individuals with heritable non-functional hOCT1 alleles, the systemic exposure was approximately two-folds higher than in individuals with functional hOCT alleles due to reduced hepatic clearance. Consequently, the drug caused more pronounced undesirable effects such as increased heart rate and blood glucose levels in OCT1-deficient individuals [123]. Further, OCT3 was proposed to mediate the uptake of cationic β_2_-agonists into their site of action (i.e., bronchial smooth muscle cells) [124]. In another study, it was demonstrated that inhibition of OCT3 in vascular smooth muscle cells, via corticosteroids, could reduce the vascular clearance of cationic β_2_-agonists and, therefore, increase their airway retention time. The authors, therefore, suggested that combining inhaled corticosteroids with β_2_-agonists may improve the pharmacologic response to the latter [30]. However, they did not discuss whether the inhibition of OCT3 in airway smooth muscle cells may instead have a negative impact on β_2_-agonists’ action by reducing their uptake into their target site.

Taken together, OCT1 and OCTN2 seem to be the main members involved in interaction with β_2_-agonists in airway epithelial cells; whereas, OCT3 is mainly involved in the uptake and clearance of β_2_-agonists into/from airway smooth muscle cells (Figure 2). By now, studies provide evidence that cationic β_2_-agonists are substrates as well as inhibitors of OCT/Ns. Based on the available data, the clinical significance of such interactions remains to be investigated.

#### 4.1.2. Interaction with Anticholinergic Drugs

Short-acting ipratropium bromide and long-acting tiotropium bromide and glycopyrronium anticholinergic drugs have been shown to be substrates of OCTs and OCTN2 [97,122,125]. OCTN2 was reported to be the predominant mediator of ipratropium bromide uptake into BEAS-2B bronchial epithelial cells [126]. In contrast, ipratropium bromide transport studies across Calu-3 monolayers revealed a net secretion sensitive to inhibition by MPP^+^ and TEA but not l-carnitine, suggesting the involvement of OCTs and ruling out any role for OCTN2 in drug flux across cell monolayers [127]. Moreover, the study revealed the presence of an active efflux mechanism that extrudes the bronchodilator to the apical chamber, which could be inhibited by MPP^+^, TEA and probenecid. The latter has been reported to inhibit multidrug-resistance-associated proteins (MRPs) [128] and organic anion transporters (OATs) [129]. However, there is no report, to our knowledge, that indicates ipratropium bromide as a substate of MRPs. OATs were undetectable in Calu-3 monolayers grown under similar condition [130]. Anions, including probenecid, have been proposed to inhibit OCTs [6] and thus might be involved in the interaction between probenecid and ipratropium bromide. However, the authors were unable to confirm this hypothesis and suggested that an apically localised OCT1 and other unidentified efflux transporters may be involved in the luminal recycling of the inhaled drug [127]. In this context, ipratropium bromide has been recognised as a substate of multidrug and toxin extrusion (MATE) transporters [131], which are reported to mediate cationic drugs’ efflux across the apical membrane of tubular renal cells and hepatocytes [132]. The authors of the above-mentioned study concluded that ipratropium absorption across respiratory epithelium is a complex process in which both passive diffusion and carrier-mediated uptake and efflux processes play a role [127]. Recently, the involvement of the amino acid transporter B^0,+^ (ATB^0,+^), which can also mediate cellular uptake of organic cations [98] and l-carnitine was discussed [133]. The transporter was taken into consideration in two recent studies in which ipratropium bromide, but not tiotropium bromide nor glycopyrrolate, was shown to inhibit OCTN2-mediated basolateral uptake of l-carnitine [38] and OCT-mediated MPP^+^ uptake [24] into 3D human bronchial epithelial cell model and Calu-3 cells. It remains to be validated whether ATB^0,+^ plays such an important role in drug disposition.

Not many ex vivo studies have been carried out to assess the interaction of anticholinergic drugs with OCT/N transporters. The uptake of ipratropium bromide, but not tiotropium bromide, was shown to be carrier mediated in lung slices obtained from drug-naïve rats [134]. In contrast, a study carried out in an isolated and perfused rat lung model suggested passive diffusion to be the main driving force for the overall absorption of ipratropium bromide across the lung epithelial barrier and that OCT/Ns play no role in the process [95]. Meanwhile, in vitro uptake experiments showed a significant role for OCTs in the uptake of ipratropium bromide into primary rat alveolar epithelial cells and into three human pulmonary epithelial cell lines (i.e., A549, BEAS-2B, 16HBE14o-).

Overall, the importance of OCT/Ns in the pulmonary disposition of anticholinergic drugs remains to some extent contradictory between in vitro and ex vivo studies. Data point towards an involvement of OCT1 and OCTN2 in the interaction with ipratropium bromide (Figure 2).

### 4.2. OCTN2 as a Target to Enhance Pulmonary Drug Delivery

Conjugation of drugs or nanodrug delivery systems with a specific transporter’s substrate to promote drug transfer across biological barriers has emerged as a strategy to improve drug delivery [135,136]. OCTN2-targeted nanodrug delivery systems have been successfully used to enhance the oral bioavailability of nanoparticles [137]. As far as the pulmonary drug delivery is concerned, Mo and colleagues synthesised a carnitine ester prodrug of prednisolone (i.e., prednisolone succinate-l-carnitine (PDSC)). The uptake of the prodrugs into BEAS-2B cells was enhanced and could be inhibited by l-carnitine, indicating it was an OCTN2-mediated process. The prodrug displayed improved duration of action with the free prednisolone being slowly released inside the cells resulting in longer suppression of LPS-induced release of IL-6 by BEAS-2B cells in vitro [138]. In a follow-up study, the asthmatic guinea pig model was treated with the prodrug and showed less severe vascular pathologies, restricted asthma induced airway thickenings and lower inflammatory cell count in bronchoalveolar fluid when compared to animals treated with unconjugated prednisolone [139]. However, a link to alterations in OCTN2 expression levels was not investigated in this model.

OCTN2 may be a potential target to enhance the pulmonary delivery of inhaled drugs to achieve a better therapeutic outcome. As mentioned before, l-carnitine itself has antioxidant properties and may confer additional beneficial effects when conjugated with inhaled drugs used for the treatment of respiratory disorders such as asthma and COPD (Figure 2).

## 5. Conclusions and Future Perspectives

Reliable epithelial cell models most closely reflecting the situation in vivo are necessary to assess the role of OCT/Ns in pulmonary (patho)physiology and drug disposition. Recent in vitro studies discussed in this review are mainly based on continuously growing cell lines such as Calu-3 [24,37,38] or A549 [40]. Comparison of OCTN2 and OCT function in respiratory epithelial cell models (i.e., Calu-3, A549 and NCl-H441) to that in human alveolar primary epithelial cells showed vast differences in kinetics and inhibitor profiles, underlining the importance of this topic [22,102]. MatTek’s EpiAirway model has been proposed as a phenotypic 3D model in which OCT/Ns were found to be functionally active at the basolateral membrane only [24,38]. These data show that an “ideal” in vitro epithelial cell model for OCTN/-based research is still elusive. To overcome some of the challenges, much effort has been undertaken to establish isolated and perfused lung models to predict in vivo drug absorption [140,141]. Furthermore, a porcine isolated and perfused lung model for pulmonary pathophysiological studies was established [142]. Porcine lungs are very similar to their human counterparts in term of size, anatomy and physiological characteristics, and this model may therefore be a promising ex vivo surrogate for inhalation biopharmaceutical investigations. When using other species, the consideration of any potential species differences in OCT/Ns’ activity between laboratory animals and human is of crucial importance. In this Special Issue, Floerl and colleagues demonstrate a good functional correlation between rat, mouse and human OCT1 in terms of interactions between a number of investigated drugs [143]. Other OCT/Ns expression and activity profiles, however, still need to be thoroughly investigated in these models.

Studies have only started considering the airway microenvironment of lung diseases on inhaled drug disposition by, e.g., modelling asthma-like conditions in vitro [37] or utilising tissue from patients [31]. The presence of mucus and airway inflammation is a crucial factor for the absorption of β_2_-agonists, e.g., salbutamol [37,144,145]. An in vitro model was developed to study mucus interaction with aerosolised drugs by applying a thin layer of porcine tracheal mucus. Results showed that mucus delayed the absorption of all inhaled tested drugs (i.e., salbutamol, formoterol and indacaterol, ipratropium bromide and glycopyrronium) but to a varying extent [145]. The thickness of the mucus layer in the model, however, was about 10-fold higher than that of the mucus film covering the airways in vivo, and thus, the observed influences are very likely overestimated. These studies considering the impact of the mucus are still in their infancy and need further technical and methodological refinement.

Following recent achievements in proteomics and microbiome analysis, which has been highly “boosted” in the last years [146,147], insights into the pathology of airway diseases become increasingly available. Several studies have shown that the airway microbiome is disturbed in patients with COPD [148,149], likely contributing to the pathology of diseases. It is yet unknown but necessary to investigate any potential change in OCT/Ns’ functions in such specific pathologic airway environments. These studies will rely on patient-derived material. By implementing, e.g., cultures of human primary 2D airway epithelial cell cultures or 3D airway spheroids [150,151], screenings of OCT/N substrates and inhibitors will greatly advance insights into responses of pulmonary drug disposition. Furthermore, this avenue of research will facilitate the analysis of effects of gene polymorphisms and gender on OCT/Ns’ expression and function (towards “individualised pharmaceutics”). Gender-related differences in the expression and activity of a number of SLC and ATP-binding cassette transporters have been observed in hepatic [152,153,154] and renal tissue [155]. Sakamoto et al. showed first evidences in their pulmonary expression profiles that females had two-fold higher levels of OCT1 and OCTN1 compared to those in males [21].

To summarise, according to the current state of knowledge, OCT/Ns have a potential pharmacological impact in the lung. The understanding of their regulation is pivotal for the development of novel inhaled drug therapies.

## Figures and Tables

**Figure 1 ijms-21-09168-f001:**
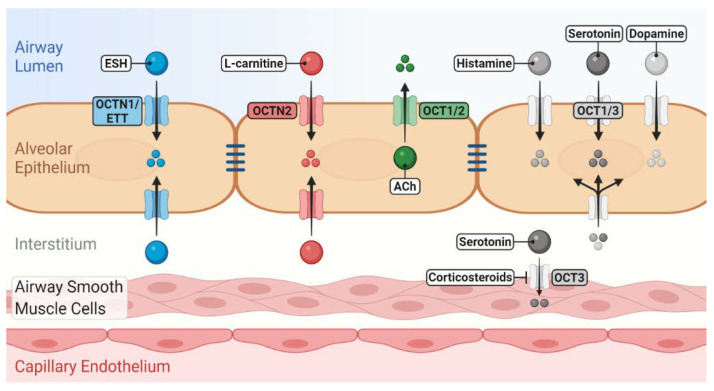
Role of novel/organic cation transporters (OCT/Ns) in pulmonary disposition of endogenous substrates. OCTN1/ergothioneine (ESH) transporter (ETT) mediates the uptake of ESH, which possesses antioxidant and anti-inflammatory properties. OCTN2 mediates l-carnitine uptake, which is essential for cellular energy production. OCTs may participate in the uptake/release of a number of neurotransmitters into/from airway epithelial and smooth muscle cells. Many of these induce bronchoconstriction, regulate mucus secretion and clearance and are linked to the pathophysiology of asthma. The localisation of OCT/Ns, whether apical or basolateral in airway epithelium, is still elusive.

**Figure 2 ijms-21-09168-f002:**
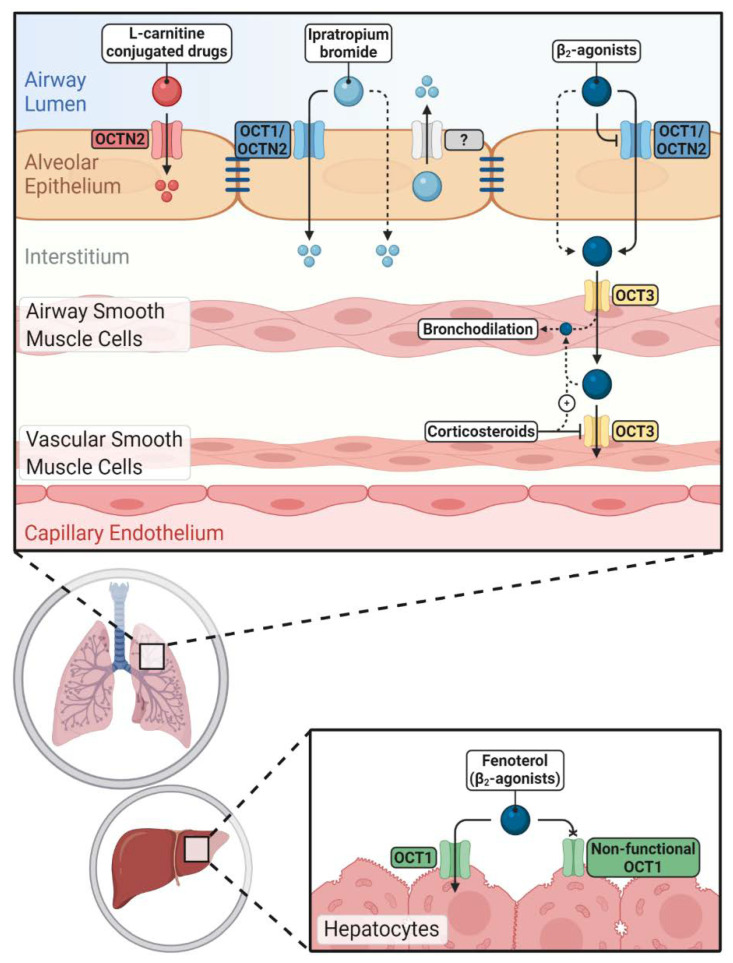
Impacts of novel/organic cation transporters (OCT/Ns) on pulmonary and systemic disposition of inhaled drugs. Inhaled cationic β_2_-agonists’ epithelial transport is mediated either by OCT/Ns (mainly OCT1 and OCTN2) or passive diffusion. Corticosteroids can increase the airway retention of β_2_-agonists by inhibiting OCT3 in vascular smooth muscle cells. Hepatic clearance of fenoterol is reduced in carriers of non-functional OCT1 alleles. Ipratropium bromide uptake via airway epithelium is more complex and OCT/Ns, unidentified efflux transporters and passive diffusion are participating. OCTN2 mediates the uptake of l-carnitine-conjugated prodrugs and nanoparticles into pulmonary epithelial cells improving pulmonary delivery of the parent drugs.

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
