# Peer review of "Organic Cation Transporters in the Lung—Current and Emerging (Patho)Physiological and Pharmacological Concepts"

_ijms, 2020, doi:10.3390/ijms21239168_

Round 1
Reviewer 1 Report
This review aims to summarize the knowledge on organic cation transporters (OCTs) including the solute carrier family (SLC) in lung homogenesis and disease, as well as their interaction with drugs and their own use a therapeutic target. This is a well written and interesting review with very nice figures on a topic that is not very prominent in lung disease (yet), I do however have some suggestions that might improve the clarity to the reader.
Major points
- It might be useful to include a scheme on the classification of the different OCTs and possibly their expression and/or function. To someone not in the field it is quite complicated to keep track of all the different abbreviations.
- The first paragraph after the introduction summarizes current data on these proteins and their function/subcellular localization. However, it is quite critical on the presented evidence (mentioning the bad quality of available antibodies, the discrepancies of different studies etc) which kind of suggests to the reader that the other points that are about to come are not to be taken seriously. I would thus suggest putting most of this paragraph to the end of chapter 3 in combination with other things that fit in there and name it critical discussions (or similar).
- Along the same line: I would actually suggest to start the review with the explanation of the different OCTs (3.2 f) and then go to the part on expression in cell/lines and disease.
- What could be explained a bit more is if there is a sort of selectivity of the different OCTs and their substrates. If there is a selectivity, where does it come from and can it maybe changed in the course of disease or an imflammatory stimuli?
Minor points
- Introduction: it might be good to quickly mention the approximate size of the proteins and their (estimated) cellular location (i.e. transmembrane protein or not) already early in the introduction, even if this is further elaborated on later.
- Page 3: I don’t see a clear reason why the commercial 3D EpiAirway epithelial cell model (MatTek 84 Corporation) is mentioned specifically here? At a first look it seems like an ALI culture that can be assembled by researchers themselves provided they have the resources. Also why would the lack of good antibodies for assessing OCTs ”constrain the development of novel in vitro cell systems”? Maybe it constrains the development of novel tools to study OCTs - but this would need to be stated more clearly.
- It might be worthwhile to quickly state the differences of the cell lines. As all of them are used often, it would be good to point out which ones are cancer cells/or transformed cells etc.
Author Response
We very much appreciate the reviewers´ constructive comments, which allowed us to significantly improve the quality of our manuscript. Below you find our point-by-point response and the respective changes which have been included in the revised manuscript.
Reviewer 1
Major points
- It might be useful to include a scheme on the classification of the different OCTs and possibly their expression and/or function. To someone not in the field it is quite complicated to keep track of all the different abbreviations.
We appreciate that the reviewer is interested in an overview table. Such tables, however, have already been published in two of our previous reviews (i.e. Salomon et al. 2012 and Nickel et al. 2016). The content of the tables is still up-to-date and, for the sake of saving space, we have expanded the Introduction and also refer the reader to our work.
- The first paragraph after the introduction summarizes current data on these proteins and their function/subcellular localization. However, it is quite critical on the presented evidence (mentioning the bad quality of available antibodies, the discrepancies of different studies etc) which kind of suggests to the reader that the other points that are about to come are not to be taken seriously. I would thus suggest putting most of this paragraph to the end of chapter 3 in combination with other things that fit in there and name it critical discussions (or similar).
We agree with the reviewer on this point. Maybe not such a good idea to start off with the bad news. Consequently, we have moved the referred paragraph to section 3.5. We thereby combined the sections on the relation of expression and functional analysis to give a clearer statement to the reader.
- Along the same line: I would actually suggest to start the review with the explanation of the different OCTs (3.2 f) and then go to the part on expression in cell/lines and disease.
We appreciate the reviewer’s suggestion and have amended the Introduction accordingly. We now provide additional information on the individual transporter proteins to make the reader more familiar with them at the beginning of the review.
- What could be explained a bit more is if there is a sort of selectivity of the different OCTs and their substrates. If there is a selectivity, where does it come from and can it may be changed in the course of disease or an imflammatory stimuli?
We thank the reviewer for raising this point. OCTs and OCTNs exhibit some specificity for their substrates (Koepsell et al. 2007). They can, however, be considered as polyspecific transporters, as they are quite promiscuous in this regard. Data about OCT/Ns in the liver and the kidneys is far more elaborated and has been reviewed by Prof. Koepsell several times in recent years (e.g. Koepsell et al. 2019, Koepsell et al. 2020). Relevant tables in our reviews from 2012 and 2016 summarise putative exogenous substrates and the respective transporting OCT in the lung. There is not enough evidence yet to draw further conclusions. To our knowledge, it has not been studied yet, if inflammatory stimuli only alter the expression levels of OCT/Ns or if the selectivity is also affected. We have discussed this topic in section 3.4. (pages 10-11, lines, 273-279) of the revised manuscript
Minor points
- Introduction: it might be good to quickly mention the approximate size of the proteins and their (estimated) cellular location (i.e. transmembrane protein or not) already early in the introduction, even if this is further elaborated on later.
This information has been included in the Introduction (page 2, lines 35-41) of the revised manuscript.
- Page 3: I don’t see a clear reason why the commercial 3D EpiAirway epithelial cell model (MatTek 84 Corporation) is mentioned specifically here? At a first look it seems like an ALI culture that can be assembled by researchers themselves provided they have the resources.
This has been amended in the revised manuscript.
- Also why would the lack of good antibodies for assessing OCTs ”constrain the development of novel in vitro cell systems”? Maybe it constrains the development of novel tools to study OCTs - but this would need to be stated more clearly.
The sentence has been rephrased in the revised manuscript (page 11, lines 295-296)
- It might be worthwhile to quickly state the differences of the cell lines. As all of them are used often, it would be good to point out which ones are cancer cells/or transformed cells etc.
We understand the reviewer´s point. According to the current structure of the paper, this would include an additional section to give more detail on epithelial cell models. Since the text is already quite long (as criticised by Reviewer 3), we have added a number of references for further reading instead in Section 3.5 (page 11, lines 296-298).
Reviewer 2 Report
Minor revision
Page 5 line 136 – 140
Can you discuss the advantage and limitations of the novel substrate detection by shifts in thermostability of the transporter protein in the presence of specific substrates? Can you comment and clarify how the transporter tandems might help to assess OCT and OCTN functional activity. Can the tandems transporter system for all transporter proteins. Tschirka et al describe the necessity of find suitable linkers, to verify the order as well as the absence of functional interference by saturation kinetics. Can you comment, if this system is applicable for different transporter for example SLC and ABC transporter as a tandem transporter system?
Page 10 section 3.4 form line 260
It would be great, if you can include functional studies with exogenous (drugs) OCTs and OCTNs compounds to a potential DDI in the lung mediated by these transporter.
Page 13 line 334 – 338
It is very important for the scientific community to discuss the limitations of in silico models, science the models like SimCyp are dependent on the data reliability generated in in vitro systems, because mostly they are performed under different experimental conditions.
Page 16 line 400
How do you explain the inhibition of the efflux mechanism that extrudes the bronchodilator drug by probenecid, which is a known inhibitor of the most organic anion transporters (OATs) and some ABC transporter. Surprisingly interact probenecid as an anion compound comparable to cationic molecule like TEA and MPP. It would be great if you discus these point.
Suggestion
The manuscript is to my opinion well done. However a summarizing table about the interaction (substrate or Inhibitor) of OCTs and OCTNs with lung specific endogenous as well as exogenous compounds, which are mentioned in the manuscript would give a great over view about the cationic transporter relevant in the lung and their interaction with different compounds
Sincerely
Yohannes Hagos
Author Response
Reviewer 2
- Page 5 line 136 – 140
Can you discuss the advantage and limitations of the novel substrate detection by shifts in thermostability of the transporter protein in the presence of specific substrates? Can you comment and clarify how the transporter tandems might help to assess OCT and OCTN functional activity. Can the tandems transporter system for all transporter proteins. Tschirka et al describe the necessity of find suitable linkers, to verify the order as well as the absence of functional interference by saturation kinetics. Can you comment, if this system is applicable for different transporter for example SLC and ABC transporter as a tandem transporter system?
We thank the reviewer for this excellent comment. We have added more details of thermostability shift assay in the updated manuscript (page 5, lines 133-136). We believe that transporter tandems are useful for the assessment of transporter activity. We have added more details explaining the assay itself and its advantages and limitations in section 3.5 of the revised manuscript (page 12, lines 309-320).
In short, a link is created between two transporters by joining their cDNAs in a single open reading frame, resulting in a 1:1 stoichiometry and thereby enabling to measure the activity of one transporter by the activity of the second one. For example, OCTN1-OCT2 tandems were used to assess the activity of OCTN1. Moreover, linking OCTN1 as a reference transporter with a number of SLC and SLCO transporters (i.e. OCT1, OCT2, OAT1, OAT3, OATP1B1 and OATP1B3), has demonstrated that transporter tandems can be a useful tool to assess the activity of transporters belonging to other families. However, there must be no functional overlap between the reference and the test transporter of the tandem which must be verified by uptake experiments with the individual unconnected transporters. In addition, suitable linker peptides should be used between the two transporters. The sequence and the length of the linkers as well as the order of transporters in the tandems may have a strong impact on transporters expression and activity and therefore must be carefully validated (Tschirka et al. 2020)
- Page 10 section 4form line 260
It would be great, if you can include functional studies with exogenous (drugs) OCTs and OCTNs compounds to a potential DDI in the lung mediated by these transporters.
We thank the reviewer for raising this interesting point. Investigations on drug-drug interactions in the lung have rarely been performed. In vitro assays, which have been mainly carried out in the pulmonary context, are not the ultimate model to test drug-drug interactions. We performed studies in NCl-H441 epithelial cells, where OCT1-mediated salbutamol uptake was affected by verapamil. However, we think that this is an interesting point and added a discussion into the paragraph.
- Page 13 line 334 – 338
It is very important for the scientific community to discuss the limitations of in silico models, science the models like SimCyp are dependent on the data reliability generated in in vitro systems, because mostly they are performed under different experimental conditions.
We thank the reviewer for bringing this point to our attention. We have amended the section and now discuss the main limitations of current in silico models in the revised version of the manuscript (page 14, lines 367-374).
- Page 16 line 400
How do you explain the inhibition of the efflux mechanism that extrudes the bronchodilator drug by probenecid, which is a known inhibitor of the most organic anion transporters (OATs) and some ABC transporter. Surprisingly interact probenecid as an anion compound comparable to cationic molecule like TEA and MPP. It would be great if you discus these point.
The reviewer raised an interesting point. We have provided more detail and discussed this point in the revised manuscript in page 17, lines 437-447.
Suggestion
- The manuscript is to my opinion well done. However a summarizing table about the interaction (substrate or Inhibitor) of OCTs and OCTNs with lung specific endogenous as well as exogenous compounds, which are mentioned in the manuscript would give a great over view about the cationic transporter relevant in the lung and their interaction with different compounds
We understand the reviewer´s point. Such tables have been published in our previous reviews (i.e. Salomon et al. 2012 and Nickel et al. 2016). Thus, we feel that including such a table is not necessary, particularly in the presence of two figures, summarising the role of OCT/Ns in disposition of endogenous compounds and inhaled drugs.
Reviewer 3 Report
Selo et al present a review of organic cation transporters (OCT) in the lung. The focus of the review is to raise awareness of these key molecules in normal and diseased lungs, as well as their potential roles in transporting pharmacological interventions to key sites within lung parenchyma (e.g., smooth muscle).
The authors present the various categories of OCT, the prototypical molecules carried by them from the epithelial surface, and then mention the problems in studying these complex transporters. They note that there is not yet a good in vitro, cell culture-based model yet to assess OCT, and also present the animal models used to assess them. The authors also remark that the expression of OCT likely varies throughout the lung, so even if one does design an in vitro model, it may be clinically relevant.
The illustrations are helpful, but the text is rather long. While the authors provide a glossary, I suggest that they consider a table that lists the various OCT, key transported molecules, key physical characteristics, and actual or potential medications transported.
Author Response
The illustrations are helpful, but the text is rather long. While the authors provide a glossary, I suggest that they consider a table that lists the various OCT, key transported molecules, key physical characteristics, and actual or potential medications transported.
We thank the reviewer for this valuable comment. Such tables have been published in our previous reviews (i.e. Salomon et al. 2012 and Nickel et al. 2016). Thus, we feel that including such a table is not necessary, particularly in the presence of two figures, summarising the role of OCT/Ns in disposition of endogenous compounds and inhaled drugs. Furthermore, we have revised the text and shortened, when possible.
Round 2
Reviewer 1 Report
The authors have answered to all my comments and have submitted an improved manuscript. I don't have more comments.